## Research Article

Magnesium ion; RNA-ion interaction; ion binding site/motif; machine learning; convolutional neural network

**Author for correspondence:**
*Shi-Jie Chen,
E-mail: chenshi@missouri.edu

# Graph deep learning locates magnesium ions in RNA

Yuanzhe Zhou[1] and Shi-Jie Chen[2]*

[1]Department of Physics and Astronomy, University of Missouri, Columbia, MO 65211-7010, USA and [2]Department of Physics and Astronomy, Department of Biochemistry, Institute of Data Sciences and Informatics, University of Missouri, Columbia, MO 65211-7010, USA

## Abstract

Magnesium ions ($Mg^{2+}$) are vital for RNA structure and cellular functions. Present efforts in RNA structure determination and understanding of RNA functions are hampered by the inability to accurately locate $Mg^{2+}$ ions in an RNA. Here we present a machine-learning method, originally developed for computer visual recognition, to predict $Mg^{2+}$ binding sites in RNA molecules. By incorporating geometrical and electrostatic features of RNA, we capture the key ingredients of $Mg^{2+}$-RNA interactions, and from deep learning, predict the $Mg^{2+}$ density distribution. Five-fold cross-validation on a dataset of 177 selected $Mg^{2+}$-containing structures and comparisons with different methods validate the approach. This new approach predicts $Mg^{2+}$ binding sites with notably higher accuracy and efficiency. More importantly, saliency analysis for eight different $Mg^{2+}$ binding motifs indicates that the model can reveal critical coordinating atoms for $Mg^{2+}$ ions and ion-RNA inner/outer-sphere coordination. Furthermore, implementation of the model uncovers new $Mg^{2+}$ binding motifs. This new approach may be combined with X-ray crystallography structure determination to pinpoint the metal ion binding sites.

## Introduction

The phosphodiester backbone of RNA carries an electronic charge per nucleotide, thus, metal ions, through binding to RNA, play a critical role in stabilizing an RNA structure. In particular, magnesium ions ($Mg^{2+}$) are essential for RNA tertiary structure folding (Pan *et al.,* 1999; Moghaddam *et al.,* 2009; Chen *et al.,* 2012; Denesyuk and Thirumalai, 2015; Welty *et al.,* 2018; Chen and Pollack, 2019), stability (Misra and Draper, 1998, 2002; Tinoco and Bustamante, 1999; Draper, 2004, 2008, 2013; Koculi *et al.,* 2006, 2007; Auffinger *et al.,* 2011; Fischer *et al.,* 2018), and function in biological processes (Pyle, 1993; Sigurdsson and Eckstein, 1995; Cate *et al.,* 1997; Hermann *et al.,* 1997; Shan *et al.,* 1999; Hanna and Doudna, 2000; Brännvall and Kirsebom, 2001; Moghaddam *et al.,* 2009; Schnabl and Sigel, 2010; Auffinger *et al.,* 2011; Denesyuk and Thirumalai, 2015). Previous experiments and theoretical studies of ion-RNA interactions have revealed some important mechanisms of specifically-bound $Mg^{2+}$, such as the observation of the cooperativity between $Mg^{2+}$ and ligand in SAM riboswitches (Hennelly *et al.,* 2012; McPhie *et al.,* 2016), and the stabilization of the group I ribozyme from the bacterium *Azoarcus* by the coordination of $Mg^{2+}$ to specific nucleotides (Rangan and Woodson, 2003; Chauhan *et al.,* 2009; Denesyuk and Thirumalai, 2015), and so forth. The results from the study of the SAM riboswitches confirm that three chelation sites of $Mg^{2+}$ in key regions of the aptamer domain can cooperate with SAM in preventing the association of the anti-terminator strand (Hennelly *et al.,* 2012), and the coarse-grained molecular simulations of the group I ribozyme indicate that the binding of the specific $Mg^{2+}$ ions correlates to the formation of the individual structural elements, and the majority of high-affinity sites are consistent with the positions of ions resolved in the crystal structure of the intron (Denesyuk and Thirumalai, 2015). The study also shows that although the principal helical domains in the *Azoarcus* ribozyme can also fold in $Ca^{2+}$, their correct relative orientation and the organization of the active site still require $Mg^{2+}$ (Denesyuk and Thirumalai, 2015). These findings definitely contribute to the crucial role of the $Mg^{2+}$ in biology.

However, experimental studies of RNA-$Mg^{2+}$ interactions are challenging. As flexible RNAs can fold to an ensemble of low-energy conformations (Sclavi *et al.,* 2005; Ritz *et al.,* 2013; Kutchko *et al.,* 2015; Woods *et al.,* 2017), experimental determination of $Mg^{2+}$ binding to RNA can be challenging because ions can bind to different RNA conformations in different ways. Furthermore, using electron density maps to distinguish $Mg^{2+}$ from water ($H_2O$) and sodium ion ($Na^+$) is challenging because they all have 10 electrons and can be distinguished only in high-resolution structures, so $Mg^{2+}$ can be easily mistaken for $H_2O$ or $Na^+$ (Nayal and Cera, 1996; Auffinger *et al.,* 2011; Zheng *et al.,* 2015; Leonarski *et al.,* 2016). Alternatively, $Mg^{2+}$

may be simply missing from crystal structures (Zheng *et al.,* 2015). A significant number of misidentified $Mg^{2+}$ binding sites can impose a strong and incorrect bias on $Mg^{2+}$ binding analysis and prediction.

In addition to the obstacles created by RNA conformational multiplicity and misidentification of $Mg^{2+}$ binding sites, a relative dearth of high-resolution structural data also imposes a barrier to the understanding of relevant biological processes that depend on RNA-$Mg^{2+}$ binding. As of January 4, 2022, 1,630 RNA-containing structures with bound $Mg^{2+}$ ions are available in the Nucleic Acid Database (Berman *et al.,* 1992; Coimbatore Narayanan *et al.,* 2013). Among these structures, 1,627 are X-ray structures, and only 1,001 are high-resolution (<3.0 Å) structures. Many of these structures come from the same molecule and organism with similar $Mg^{2+}$ binding sites, and thus are effectively redundant. Experimental determination of high-resolution structures is time-consuming, which makes computational prediction of $Mg^{2+}$ binding a much desired complementary approach. The growing number of experimentally solved RNA structures motivates us to take advantage of the increasing amount of experimental information by developing a data-based method to predict and analyse the interactions between RNA and $Mg^{2+}$ ions.

During the last few years, researchers have developed several novel approaches to predict RNA-metal ion binding sites. We can categorize these modelling efforts into physics-based approaches and knowledge-based approaches. Physics-based methods, such as all-atom molecular dynamics (MD) simulations (Hanke and Gohlke, 2015; Bergonzo *et al.,* 2016; Lemkul *et al.,* 2016; Bergonzo and Cheatham, 2017; Casalino *et al.,* 2017; Fischer *et al.,* 2018; Hayatshahi *et al.,* 2018; Mamatkulov and Schwierz, 2018; Cruz-León *et al.,* 2021; Grotz *et al.,* 2021), Brownian dynamics simulations (Hermann and Westhof, 1998; van Buuren *et al.,* 2002), Poisson–Boltzmann (PB)/generalized Born (GB) models (Misra and Draper, 2000; Onufriev *et al.,* 2000; Burkhardt and Zacharias, 2001; Tolokh *et al.,* 2018; Onufriev and Case, 2019), and statistical mechanical models (Tan and Chen, 2005; Hayes *et al.,* 2015; Sun and Chen, 2016), explicitly consider physical energetics and dynamics for RNA-ion binding. In addition to the methods mentioned above, hybrid quantum mechanics/molecular mechanics (QM/MM) simulations and density functional theory (DFT) have been extensively used to study RNA-$Mg^{2+}$ interactions and the roles of $Mg^{2+}$ in various ribozyme activities such as the self-cleavage of HDV ribozyme (Mlỳnskỳ *et al.,* 2015; Thaplyal *et al.,* 2015), the hammerhead ribozyme (Chen *et al.,* 2017), and the *glmS* ribozyme-GlcN6P cofactor complex (Zhang *et al.,* 2016), in the splicing mechanism of group II introns (Casalino *et al.,* 2016), and in the stabilization and fine-tuning for noncanonical base pairing geometries that are otherwise unstable in the absence of $Mg^{2+}$ binding (Halder *et al.,* 2017, 2018). However, given the complex physical interactions considered, these approaches are often computationally demanding with various levels of success. Knowledge-based methods, such as FEATURE (Banatao *et al.,* 2003) and MetalionRNA (Philips *et al.,* 2011), on the other hand, rely on information extracted from experimentally determined structures. Such methods are usually much less computationally demanding than physics-based approaches, but the inability of taking long-range, many-body physical features into consideration limited the accuracy of these models. For example, FEATURE (Banatao *et al.,* 2003), a Bayesian-inference-based statistical model, can predict the magnesium ion-binding sites in RNA structures with the prior knowledge of the binding/non-binding environments (i.e. microenvironments) learned from the

dataset. The microenvironment is essentially defined by a collection of physical and chemical features at different levels of detail from atom, chemical group, and nucleotide-residue, to secondary structural levels – that exhibit statistically significant differences between the distributions of the known ion-binding sites and the control non-binding sites. When given a query region in a new structure, the Bayesian-inference-based scoring function can rank the sites in the query region based on the prior knowledge of the features learned from the training set. MetalionRNA (Philips *et al.,* 2011) uses a representative set of 113 crystallographically determined structures to derive statistical potentials for $Na^+$, $K^+$, and $Mg^{2+}$ ions. The model evaluates the three-body anisotropic contact frequencies between metal ions and a set of predefined covalently bonded RNA atom pairs that are known to make the strongest contributions to metal ion binding. The model then transforms the contact frequencies into statistical potentials through the inverse Boltzmann law. Given a new structure, MetalionRNA scores every grid point in the space according to statistical potentials derived from the observed contact frequencies in the training set. These scores are used to predict the final binding sites.

However, there are two main drawbacks to these approaches: the feature design requires excessive manual interventions and the scoring functions fail to take many-body effects into consideration. First, both approaches require a set of manually engineered features/atom pairs to encode the interactions. The choice of these features can be crucial and would certainly affect the performance of the model. For example, the existence of the redundant features could easily introduce bias to the prediction. Second, the fact that both approaches employ a scoring function as an additive sum of the contributions from each individual feature/atom pair implies that the scoring function does not account for many-body correlations between the different contributions.

Here, we present MgNet, a variant of the regression convolutional neural networks (Adhikari *et al.,* 2017; Li *et al.,* 2018) with residual shortcuts (He *et al.,* 2016), which uses experimental structural data to predict metal ion binding sites. In contrast to the aforementioned previous knowledge-based approaches, CNN models excel at pattern recognition by using convolutional operations to combine correlated data and identify underlying trends. It does not require manually engineered features or predefined functional forms for the scoring function, and the underlying important features and the correlations between them can be learned from the data automatically during the training process.

## Materials and methods

### Curating the data sets

In order to generate a suitable collection of images, we use a set of 177 crystallographically determined structures containing RNA and $Mg^{2+}$ ions in the Protein Data Bank (Berman *et al.,* 2000), including protein-RNA and DNA-RNA complexes. These 177 structures are selected according to the following criteria. First, RNA structures containing $Mg^{2+}$ ions were gathered from the Protein Data Bank (Berman *et al.,* 2000). A structure might be determined from different labs for the same RNA, a mutant, or a ligand-bound complex. As a result, for a given RNA, the Protein Data Bank may contain more than one structure file. To remove structure redundancy, we cluster the $Mg^{2+}$-containing RNA structures based on the nonredundant RNA structure datasets ((Leontis and Zirbel, 2012) version 3.54), and select one structure from each

sequence/structure equivalence cluster. Due to computational limitations, for large RNAs, we select only one 16S rRNA (~1,500 nucleotides). Because the resolution of crystallographic structures is a key factor for the accurate determination of the identity and position of $Mg^{2+}$, we keep only structures with a resolution of 3 Å or better. While allowing curation of a training set with sufficient data, this resolution cut-off serves to exclude structures that may misidentify $Mg^{2+}$ binding sites. For structures with multiple models, we use the first model, and for residues with more than one alternative conformation, we use the first variant. In order to apply a five-fold cross-validation evaluation, the 177 RNA-containing structures ("general set") are randomly divided into five subsets (Supplementary Table S1).

### Outlining the methods

While normal CNNs read 2D images as input, our MgNet reads "3D images" that contain the local environment of the binding and non-binding sites as input. These "3D images" provide electrostatic and 3D-shape (RNA volume) information that determines the interaction between RNA and metal ions (Fig. 1a). Molecular modelling software, such as UCSF Chimera (Pettersen *et al.*,

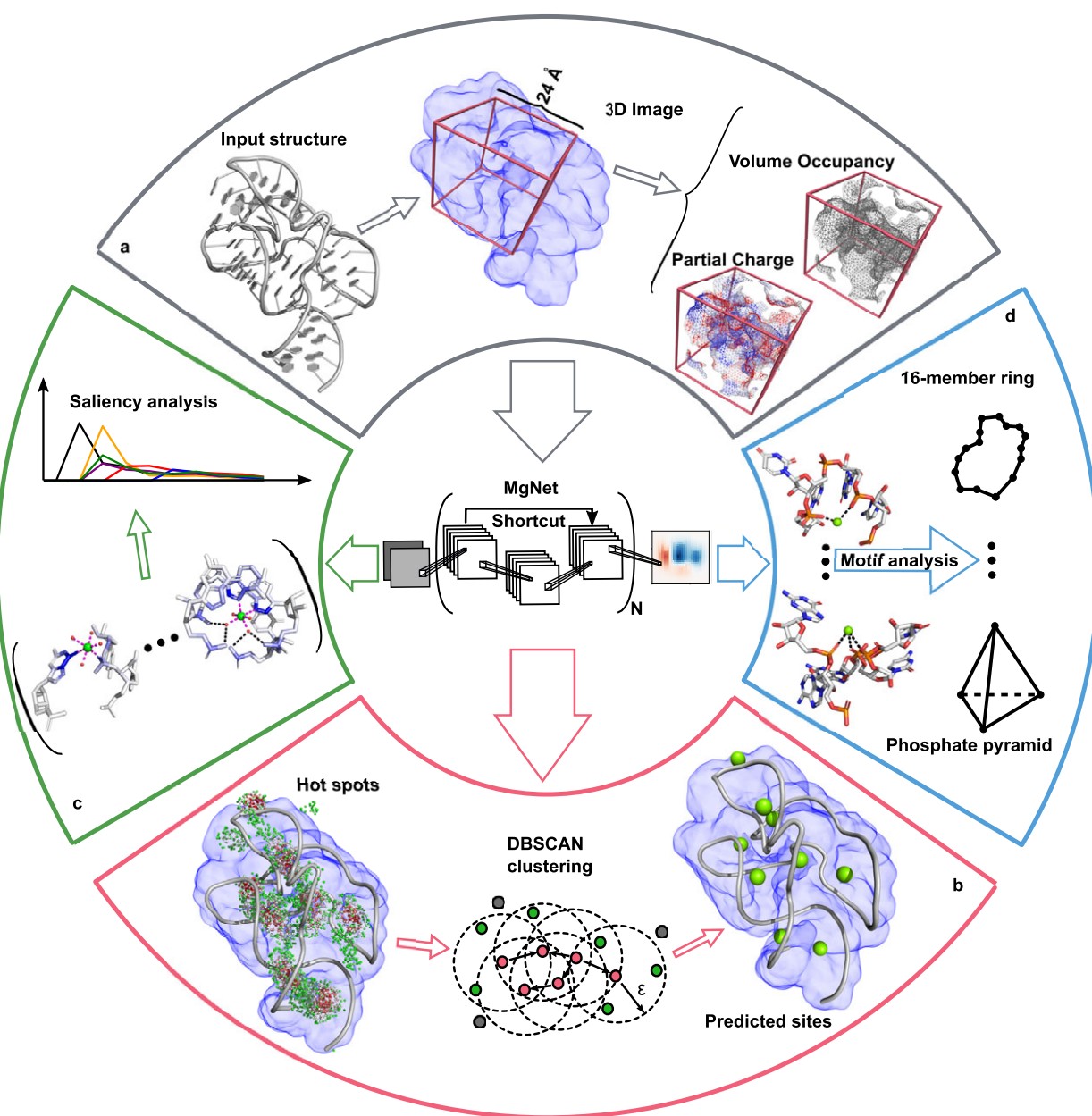

**Fig. 1.** The MgNet workflow (*a,b*) and applications (*c,d*). (*a*) The MgNet workflow begins with input of the 3D structure of a RNA. 3D image is taken from a 24 × 24 × 24 Å cubic box centred at each given nucleotide and is used to capture the electrostatic and 3D-shape information for the binding and non-binding sites. The MgNet accepts the input images and can be used to perform: (*b*) $Mg^{2+}$ binding site prediction. The hot spots (left, with decreasing probability from red to green) were collected, sorted, and clustered into final predicted binding sites (right, green spheres); (*c*) Saliency analysis. MgNet can be used to reveal the most important coordinating RNA atoms by calculating the radial saliency distributions of different atom types around the bound ion; (*d*) Binding Motif analysis. Statistics of the configurations of the coordinating atoms around the binding sites predicted by MgNet lead to newly discovered binding motifs.

2004), High-Throughput Molecular Dynamics (HTMD) (Doerr *et al.*, 2016), Visual Molecular Dynamics (VMD) (Humphrey *et al.*, 1996), Biopython (Cock *et al.*, 2009), and AutoDockTools4 (Morris *et al.*, 2009), is used to compute the partial charges of the RNA atoms and perform the voxelization for the graphical convolutional neural network. With the generated images, the MgNet predicts $Mg^{2+}$ ion probability distribution around the RNA (Fig. 1*b*). To identify $Mg^{2+}$ binding sites from the predicted ion probability distribution, we use the DBSCAN (Ester *et al.*, 1996) method to cluster the ion binding sites of probability maxima. Within each high-probability region, k-means clustering is used to find the representative points of the region. These representative points are chosen as the predicted ion sites and ranked based on the sum of the probabilities of the points within the corresponding cluster. In this work, we mainly use true positive rate (TPR) and positive predictive value (PPV) to measure the predictive power of the model. TPR (PPV) is the ratio between the number of the correctly predicted ion binding sites out of the experimentally observed (theoretically predicted) bound ions. Generally speaking, although one may alter TPR and PPV by adjusting the definition of the "correctly" predicted sites, these two metrics are often antagonistic to each other except for a perfect model. In practice, increasing the number of the predicted sites usually improves the TPR but in the meantime, causes the degradation of the PPV, and vice versa. Thus TPR and PPV together can provide an overall measure of the performance of the model.

We also aim to uncover physical insights from the neural network "black box". Specifically, we perform saliency calculation

(Fig. 1*c*) and motif analysis (Fig. 1*d*). From the gradients of the predicted scores with respect to the input image pixels (saliency values), the saliency analysis (Smilkov *et al.*, 2017) identifies the most sensitive pixels in the input image whose small variations cause substantial changes in the output result. The saliency technique allows us to uncover the critical RNA atoms that most sensitively determine $Mg^{2+}$ binding. Furthermore, from a thorough investigation of the configurations of RNA atoms around a bound $Mg^{2+}$ ion, we uncover $Mg^{2+}$ binding motifs. Here an $Mg^{2+}$ binding motif is defined as a recurring pattern of coordinating RNA atoms (i.e. geometric arrangement and atom type of the coordinating atoms) surrounding a bound ion.

## Results

### Evaluating MgNet performance through cross-validation

We carry out five-fold cross-validation on the general set with 177 RNA-$Mg^{2+}$ complex structures. For each cycle, we use one of the subsets for testing and the other four for training the MgNet model. The cross-validation approach ensures the complete sampling of the entire data sets while keeping test and training sets not overlapping in the same cycle. As shown in Fig. 2*a*, the small fluctuations among TPR (PPV) values across five folds indicate the robustness of the MgNet model. As a summary, for the 177 RNA-$Mg^{2+}$ complex structures, there are 1,407 experimentally determined $Mg^{2+}$ binding sites, MgNet predicts 1,863 $Mg^{2+}$ binding sites, among which 661 $Mg^{2+}$ binding sites (coordinates) are

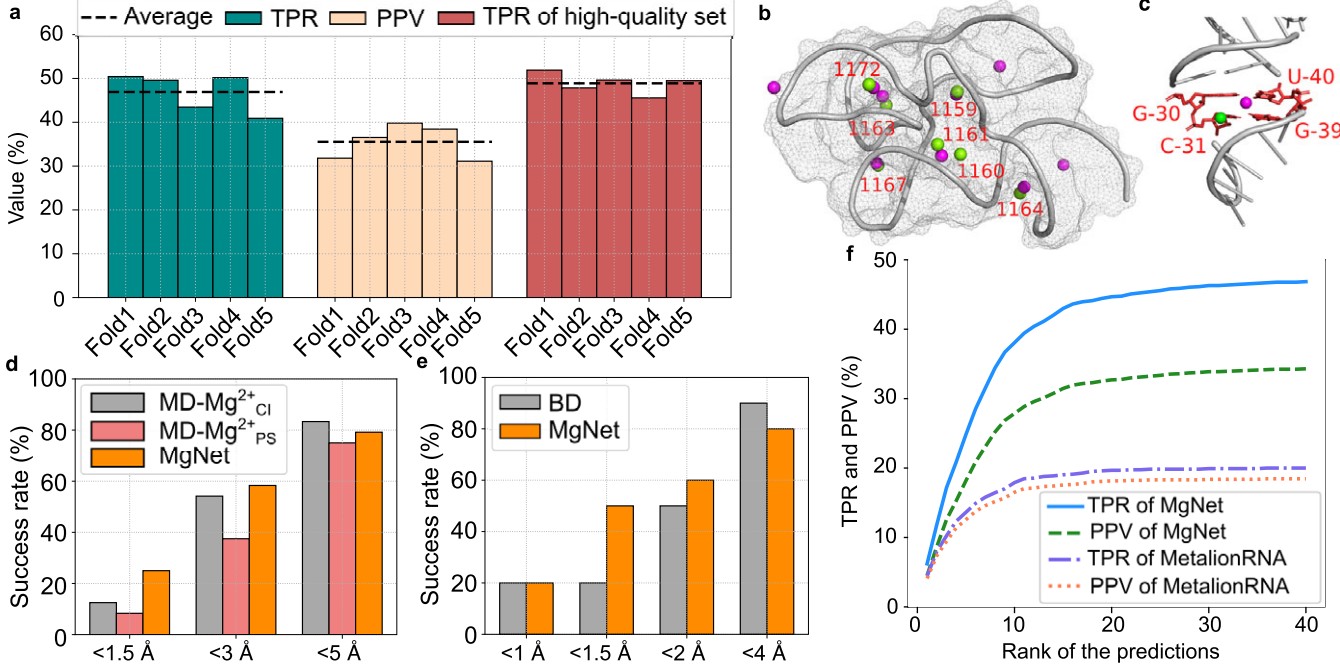

**Fig. 2.** Investigation of MgNet performance and comparison between MgNet and other methods. (*a*) The TPR and PPV values of the MgNet model for cross-validation on both the general and high-quality set. Values are obtained from validation results, PPV values on the high-quality set are not shown. (*b,c*) Example of MgNet-predicted (magenta spheres) *versus* experimentally determined (green spheres, labelled with residue identifiers) $Mg^{2+}$ ion sites in (*b*) 58 nt fragment of *Escherichia coli* 23S rRNA (PDB ID: 1HC8) and (*c*) the anticodon loop in tRNA$^{Asp}$. The predicted site in (*c*) is shifted upward toward the $G_{30}·U_{40}$ wobble pair. Four residues shown in red are labelled with the residue names and residue sequence numbers. (*d,e*) Comparison of the success rates between the MgNet and molecular dynamics (MD) and Brownian dynamics (BD) simulation-based methods for various RMSD cut-offs. The test sets contain seven and three RNA structures for MD-based and BD-based method, respectively. Two different system conditions were used in MD-based method, with $Mg^{2+}$ as the counterion (CI) ($Mg_{CI}^{2+}$) only and with the physiological salt (PS) concentration $Mg_{PS}^{2+}$ ($Mg^{2+}$ counterions and 0.15 M NaCl). (*f*) Comparison between MetalionRNA (Philips *et al.*, 2011) and MgNet on the general set. The horizontal axis represents the rank of the predictions, where *n* on the axis means the top-*n* predictions is used for each RNA, and the vertical axis represents the corresponding TPR and PPV values for the top-*n* predictions. The cut-off RMSD for a correct hit is 3 Å. Additional information can be found in Supplementary Tables S4–S8.

within 3 Å from the experimental results. Statistically speaking, the test result implies that the MgNet model is able to identify nearly half (661/1,407) of the true $Mg^{2+}$ binding sites with high accuracy. Details can also be found in Supplementary Dataset S1.

In addition to the above cross-validation, we also employ the five-fold cross-validation to validate the MgNet model on another dataset with 1,974 high-quality $Mg^{2+}$ binding sites clustered from the MgRNA benchmark set (Zheng *et al.,* 2015) (see Supplementary Information). The purpose of MgNet computation/validation with the high-quality set is to validate the robustness of the MgNet model against the different datasets. However, sites in the high-quality set were chosen from experimentally determined RNA structure, where many (experimentally derived) sites not included in the set could be close to the included ones. This would make the "false positive (FP)" of the prediction ambiguously defined (see Supplementary Figure S2). For this reason, we only use the TPR (equivalent to success rate) to evaluate the performance. The results are shown in Fig. 2*a* and Supplementary Table S3. The similar TPR results for both the general set and the high-quality set suggest a consistent performance of MgNet.

### MgNet and MetalionRNA

By comparing MgNet to the knowledge-based method MetalionRNA (Philips *et al.,* 2011), we assess the performance of the CNN approach. Following the previous studies (Banatao *et al.,* 2003; Philips *et al.,* 2011), we first investigate the MgNet predictions on the 58 nt fragment of *Escherichia coli* 23S rRNA which contains seven $Mg^{2+}$ ions in the crystal structure (PDB code 1HC8, Fig. 2*b*). As also shown in Supplementary Table S4, MgNet and MetalionRNA can both identify all the seven $Mg^{2+}$ ions within the top-12 and top-29 ranked predictions with an accuracy of 0.5–2.3 Å and 0.6–3.8 Å, respectively.

For a more comprehensive comparison, we use TPR and PPV to evaluate the performance of MetalionRNA (Philips *et al.,* 2011) on our cross-validation dataset. Fig. 2*f* shows the distributions of the TPR and PPV values from the MgNet model and the MetalionRNA web server on the 176 RNA-containing structures over the number of top predictions. It can be seen that the curves diverge quickly with the increase in the rank of the predictions, suggesting that MgNet has a notably better success rate in predicting the experimental ion binding sites.

### MgNet and a molecular dynamics (MD) simulation model

Although several physics-based methods have been developed to investigate the metal ion-RNA interactions, most methods focus on the dynamics or statistical properties instead of the ion binding sites. As suggested by Fischer *et al.* (2018), an MD method with explicit water can be applied to characterize $Mg^{2+}$ distributions around folded RNA structures and to predict $Mg^{2+}$ positions. In the study (Fischer *et al.,* 2018), seven RNA structures containing $Mg^{2+}$ ions are selected as the target system in MD simulation. In order to test whether MD simulation can recover the experimental binding sites, ions are initially randomly placed in the simulation box. The predicted ion positions are determined by the occupancy of $Mg^{2+}$ during the simulation using the software MobyWat (Jeszenői *et al.,* 2015, 2016).

To compare the MgNet predictions with the MD simulation results for the seven RNA structures, we use a five-fold cross-

validation procedure. We use the same five subsets of RNA structures generated from the general set. For each subset, we remove possible duplicate RNA structures of the seven test structures. This step results in the removal of RNA structures with PDB codes 1D4R, 1Y95, and 4FRG, leaving 174 remaining RNA structures. We then perform the five-fold cross-validation for the five (modified) subsets. Finally, we use each trained model to predict the $Mg^{2+}$ binding sites for the seven test RNA structures. The success rates of MgNet and MD simulation methods are shown in Fig. 2*d*. By investigating the details of the predictions (Supplementary Tables S5–S7), we found the MgNet model gives overall better predictions than the MD simulations for identifying the locations of the bound ions with small RMSD cut-off. The difference between the MgNet and the MD simulation results is due to the following reasons. First, the RNA structures used in MgNet training are mainly crystal structures, thus the interaction patterns learned by MgNet may not be ideal for NMR solution structures, which causes slightly worse results for 2MTK (PDB ID), an NMR solution structure. Second, MD simulations for ions directly bound to RNA may suffer from the incomplete sampling problem due to the high barrier for $Mg^{2+}$ dehydration.

### MgNet and a Brownian dynamics (BD) simulation-based method

In Brownian dynamics (BD) simulations (Hermann and Westhof, 1998), diffuse cations move under the influence of random Brownian motion in the electrostatic field and the metal ion binding sites are identified by analysing the trajectories of positively charged test particles. Previous BD simulations have shown the ability to identify $Mg^{2+}$ binding sites in the crystal structures of loop E of bacterial 5S rRNA (PDB code: 354D), tRNA$^{Phe}$ (PDB code: 4TRA) and tRNA$^{Asp}$ (PDB code: 3TRA). To compare MgNet with the BD simulations, we use the aforementioned five-fold cross-validation procedure with the test RNA structures removed from the general set. The resultant dataset contains 175 RNA structures.

As shown in Fig. 2*e* and Supplementary Table S8 for the comparison between the BD simulations and our MgNet models, overall both BD simulations and MgNet show good performance for the tested RNA structures. However, there exist two notable differences between the predictions from the two approaches. Several trained models of MgNet fail to predict the binding sites within 10 Å from the experimental sites for $Mg^{2+}$ ion A-76 (354D) and ion A-80 (4TRA). One predicted site within 10 Å is captured for ion A-76, and the RMSD of the MgNet-predicted ion A-80 is larger than that of BD simulation. For ion A-76 of 3TRA, the crystal structure of tRNA$^{Asp}$ contains a single $Mg^{2+}$ located in the anticodon loop at the $C_{31} \cdot G_{39}$ base pair (Hermann and Westhof, 1998). Both BD simulations and MgNet-predicted ion sites are within ~5 Å from the site in the crystal structure, and both are shifted upward in the anticodon stem towards the $G_{30} \cdot U_{40}$ wobble pair (Fig. 2*c*). This shifted ion binding pattern is similar to the experimentally found metal ion binding site at G·U pairs in the crystal structure of P4–P6 of group I intron (Hermann and Westhof, 1998). The result might indicate a delocalized binding of metal ions in the anticodon loop of tRNA$^{Asp}$ as suggested by Hermann and Westhof (1998). As for ion A-80 of 4TRA, the predicted site deviates from the experimental site possibly because this particular ion is in close contact with a non-standard residue Wybutosine (yw). We note that

$Mg^{2+}$ binding to one or more non-standard residues is not common in our training set, thus the predictions of MgNet for such cases may be less reliable.

## MgNet-saliency analysis for metal ion binding sites

In machine learning, a large saliency value means that a slight change in the corresponding input feature causes a large change in the prediction score. Therefore, saliency analysis can identify the key physical features that most sensitively determine ion binding. In MgNet, from each input 3D image, the convolutional network predicts a 3D matrix where a matrix element p($i, j, k$) is the probability of finding a bound ion at the grid site ($i, j, k$). From the gradients of the predicted ion distribution with respect to the input pixels of the images of the target binding site, the saliency analysis identifies the RNA atoms and the physical attributes that determine ion binding.

Eight representative binding sites of distinct motifs from the previous survey (Zheng *et al.,* 2015) are picked from the general set. Six cases (Fig. 3*a*–*f*) involve inner-sphere interactions with RNA atoms, while the rest (Fig. 3*g*–*h*) interact with RNA atoms through outer-sphere hydrogen bonds (mediated by water molecules). Several motifs share geometrical similarities. Through the juxtaposition of two different strands or two distant segments of the same strand, the "Magnesium clamp" (Ennifar *et al.,* 1999; Petrov *et al.,* 2011) and "Y-clamp" (Zheng *et al.,* 2015) use the bridging capability of phosphates to stabilize these close interactions, very much similar to the disulphide bonds in proteins. The "U-phosphate" (Zheng *et al.,* 2015) and "G-phosphate" (Klein *et al.,* 2004) both require the coordination of phosphate oxygen and nucleobase oxygen. The more complicated motifs, "Purine N7-seat" (Zheng *et al.,* 2015), "G-G metal binding site" (Correll *et al.,* 1997), and "Triple G motif" (Tinoco and Kieft, 1997), contain complex water-mediated coordination.

The saliency value of a particular atom reflects the sensitivity of the predicted ion density with respect to this particular atom, namely, a small change in the pixel values (physical attributes) of the blue atoms shown in the figure would markedly alter the predicted ion (probability) density. Therefore, saliency analysis for the above examples can uncover important atoms that are critical for the stabilization of magnesium ions at the binding site. As shown in Fig. 3, atom saliency values for the two input channels (volume occupancy and partial charge) indicate specific coordinating atoms as the important factors in determining $Mg^{2+}$ binding sites. Note that in Fig. 3*a*, two of the important phosphate oxygen atoms (OP1 of A34 and OP2 of G46) in the opposite direction have a large saliency value (a darker colour), suggesting a critical role of these atoms in ion binding. Indeed, there exists another $Mg^{2+}$ ion that binds in the nearby location (shown as a cyan sphere). The coordinating atoms (connected through dashed lines) have relatively large saliency values, indicating their importance in $Mg^{2+}$ ion binding. Indeed, as shown in Supplementary Table S9, for the motifs shown in Fig. 3, all of the binding sites can be successfully predicted by the MgNet model for the original RNA structures. However, after removing the coordinating atoms, MgNet fails to find the correct binding sites for six cases. The result again supports the important role of the identified RNA atoms.

To further investigate the spatial distribution of the RNA atoms around the bound ions, we classify four types of RNA atoms (Zheng *et al.,* 2015): (i) $O_{ph}$, phosphate oxygen (OP1/OP2); (ii) $O_r$, oxygen in ribose (O2'/O4') or oxygen bridging phosphate and ribose (O3'/O5'); (iii) $O_b$, nucleobase oxygen and (iv) $N_b$, nucleobase nitrogen,

where the last two types ($O_b$ and $N_b$) are further divided into subtypes according to the nucleotide type (purine or pyrimidine), resulting in overall six types. Then, we use the radial distribution function to quantify the spatial frequency and saliency distribution of the different atom types around a bound ion (see Fig. 4).

The contact frequency distribution, as shown in Fig. 4*a*, shows two characteristic peaks at ~2.3 and ~ 4.3 Å, corresponding to inner-sphere and outer-sphere coordinations, respectively. The peak at ~2.3 Å for $O_{ph}$ indicates that $O_{ph}$ is the most abundant inner-sphere coordinating atom, and the peak at ~ 4.3 Å comes from the water-mediated coordination. For purine-$N_b$, we find multiple nitrogen atoms in guanine/adenine residue that are spatially correlated, which explains the peaks around ~ 4.3 and ~ 6.3 Å. We note the distribution curves become flat as distance increases, reflecting the relative abundance of these atom types in our cross-validation set.

The radial distributions of saliency values for volume occupancy and partial charge channels, as shown in Fig. 4*b*,*c*, are peaked at smaller radial distances than the contact frequency distribution in Fig. 4*a*. The shift in the peak positions is because $Mg^{2+}$ is more sensitive to the closer coordinating atoms. Furthermore, the saliency peaks of the different atom types in the partial charge channel are higher than those in the volume occupancy channel, except for $O_r$. The result suggests that $Mg^{2+}$ binding sites are more sensitive to the partial charges of the coordinating atoms than the occupancy of RNA atoms. The abnormal behaviour of $O_r$ may be caused by its spatial correlations with $O_{ph}$. In the volume occupancy channel, $O_{ph}$ and $O_r$ often appear together as coordinating atoms, thus showing similar peaks in the saliency distribution. In contrast, in the partial charge channel, the partial charge of an $O_r$ is less than that of an $O_{ph}$ and thus shows a lower peak (weaker sensitivity).

To further identify the critical atoms, we investigate the radial frequency distribution and the relative saliency distribution of each individual atom. The trend of the radial frequency distributions of the representative atoms within 3 Å (Fig. 4*d*) are very similar to the atom-type distributions (Fig. 4*a*), where the normalized radial frequency distributions (Fig. 4*d*) are roughly twice as large due to the fact that $O_{ph}$ contains two phosphate oxygen atoms (OP1 and OP2). The similar distributions suggest that these representative atoms are indeed the dominant inner-sphere coordinating atoms for each RNA atom type. Thus, the saliency distributions (Fig. 4*e*,*f*), which are dominated by RNA atoms with close contact with $Mg^{2+}$, also show similar trends as in Fig. 4*b*,*c*.

## Identifying novel $Mg^{2+}$ binding motifs

MgNet leads to two novel $Mg^{2+}$ binding motifs that have not been reported (Zheng *et al.,* 2015). Typical $Mg^{2+}$ can coordinate with six atoms forming octahedral geometry, these coordinating atoms are usually electronegative oxygen/nitrogen atoms from either water molecules or RNA molecules. In this study, since MgNet does not treat outer-sphere coordination (i.e. interactions mediated by water molecules), we focus on motifs involving inner-sphere coordination with RNA atoms.

For the 373 representative sequences/structures (Supplementary Information), MgNet predicts 1,137 binding sites with inner-sphere coordination, among which 313 are previously reported binding motifs and 654 are inner-sphere coordination binding sites with a single coordinating RNA atom. For single atom-coordinated sites, the bound $Mg^{2+}$ ions could be partially dehydrated and it is possible that some of these sites involve outer-sphere $Mg^{2+}$ binding motifs

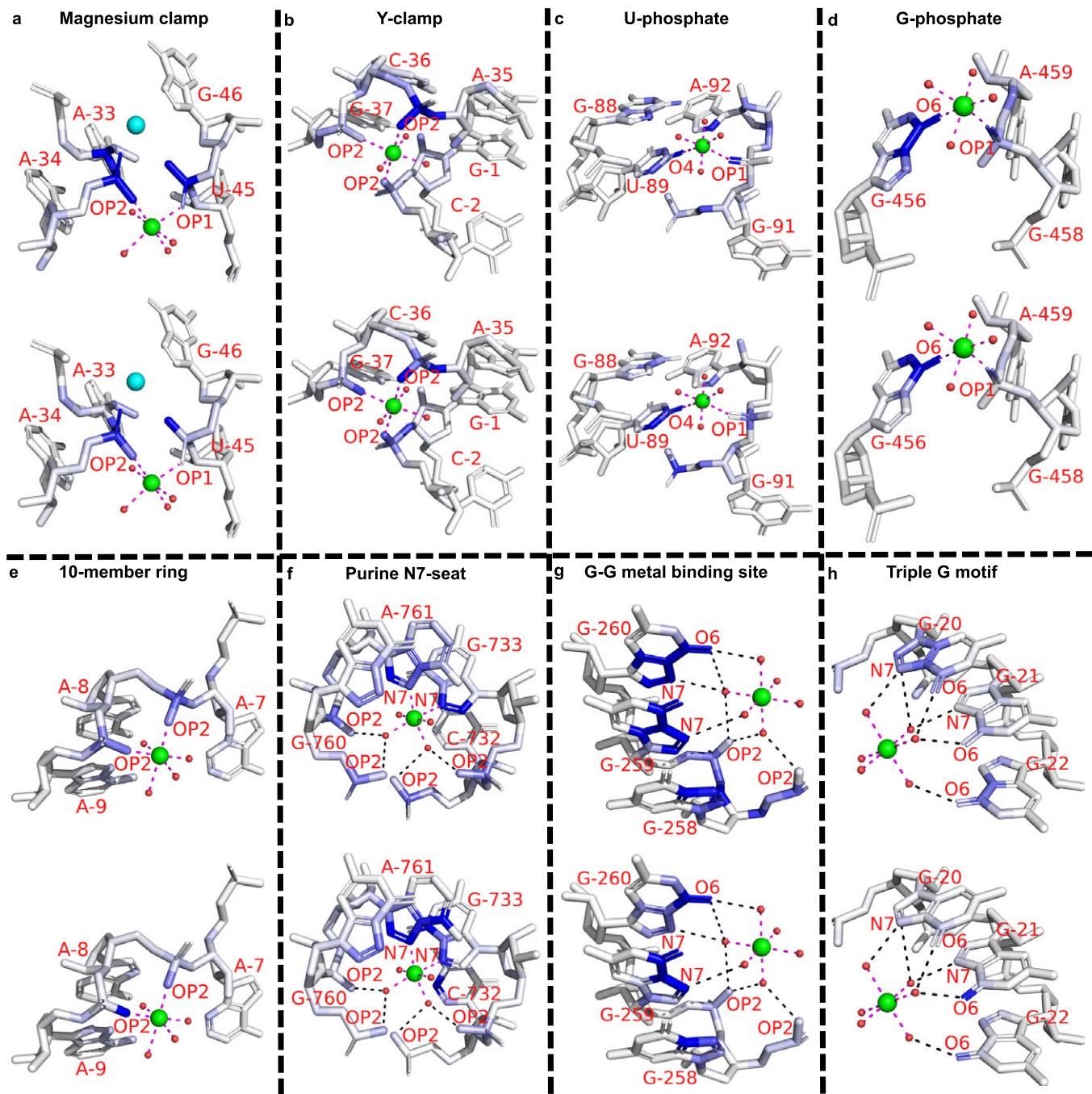

**Fig. 3.** Example of saliency calculation for eight binding motifs. These motifs differ by the type of ion coordination (i.e. inner-sphere or outer-sphere coordination), the number and type of the coordinating atoms, and the geometry of the coordination. Saliency values are calculated for eight binding sites: (*a*) 3Q3Z-V85; (*b*) 2Z75-B301; (*c*) 2YIE-Z1116; (*d*) 1VQ8–08004; (*e*) 3DD2-B1000; (*f*) 2QBA-B3321; (*g*) 4TP8-A1601; (*h*) 3HAX-E200, and two input channels: volume occupancy (top) and partial charge (bottom). Experimentally determined positions of $Mg^{2+}$ cation are indicated by green spheres, oxygen atoms in water molecules are shown in small red spheres. Direct coordination (inner-sphere coordination) are shown as magenta dashes, and indirect coordination (outer-sphere coordination, i.e. mediated by water molecules) are shown as black dashes. Residues and coordinating atoms other than oxygen of water molecules are labelled with red text. One extra $Mg^{2+}$ in (*a*) is shown as a cyan sphere. The saliency values of RNA atoms are shown in the blue scale, where the atoms with larger saliency values are shown in a darker blue colour.

with water-mediated outer-sphere interactions. However, our current MgNet model is unable to identify the position of the coordinating water molecules thus $Mg^{2+}$ coordinated by a single RNA atom is not considered as a robust motif in this study. From the remaining 170 sites with inner-sphere coordination, we identify two new binding motifs, namely, the "16-member ring" and "Phosphate pyramid" (Fig. 5*a*,*b*).

Furthermore, we compute the relative abundance of the previously reported binding motifs and the newly found ones for both the MgRNA benchmark set (Zheng *et al.,* 2015) and the general set (Fig. 5*c*). The MgRNA benchmark set contains comprehensive high-quality $Mg^{2+}$ binding sites and was previously used to identify $Mg^{2+}$ binding motifs (Zheng *et al.,* 2015). For previously reported inner-sphere motifs, only top-5 abundant motifs are plotted. The

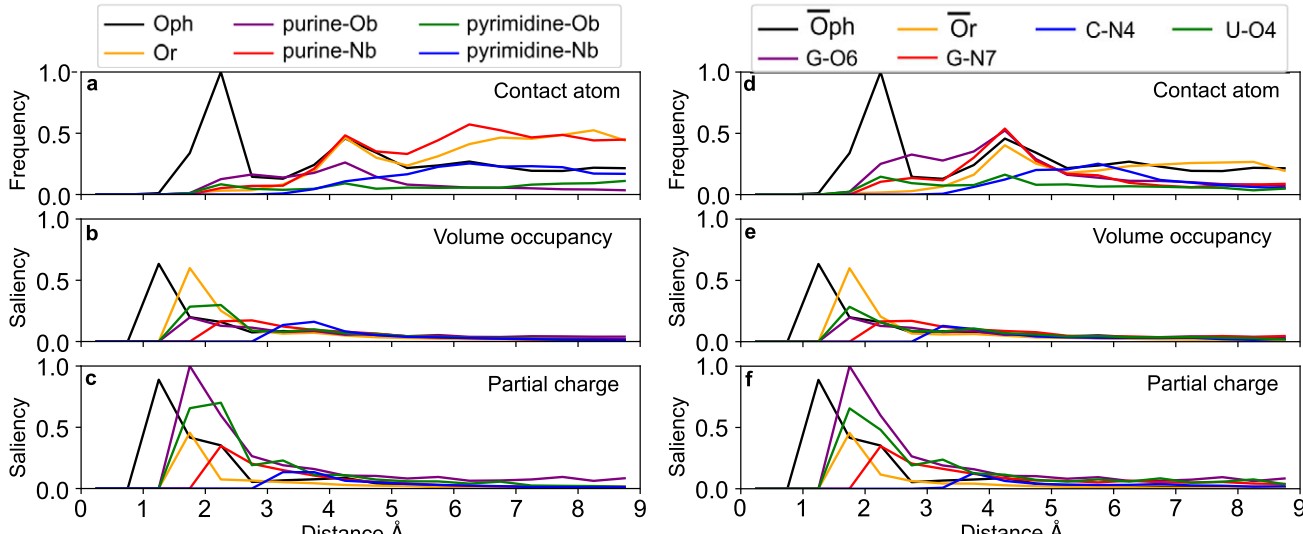

**Fig. 4.** Radial frequency distributions and relative saliency distributions of different (*a–c*) atom types and (*d–f*) representative atoms around the correctly predicted Mg²⁺ ion sites. The figure shows the contact radial frequency distributions (*a,d*), the relative saliency distributions for the volume occupancies (*b,e*) and the partial charges (*c,f*), respectively. The frequencies and saliency values are normalized to the [0, 1] range. In (*d–f*), only the representative atom of each atom type is shown (with the same colour as the corresponding atom type in (*a–c*)). $\bar{O}_r$ is the average of two sugar oxygen atoms (O3' and O5') due to the similar radial frequencies and relative saliency distributions, and $\bar{O}_{ph}$ is the average of the two phosphate oxygen atoms OP1 and OP2. The representative atoms are chosen by selecting the most abundant atom for each atom type. Details can also be found in Supplementary Information.

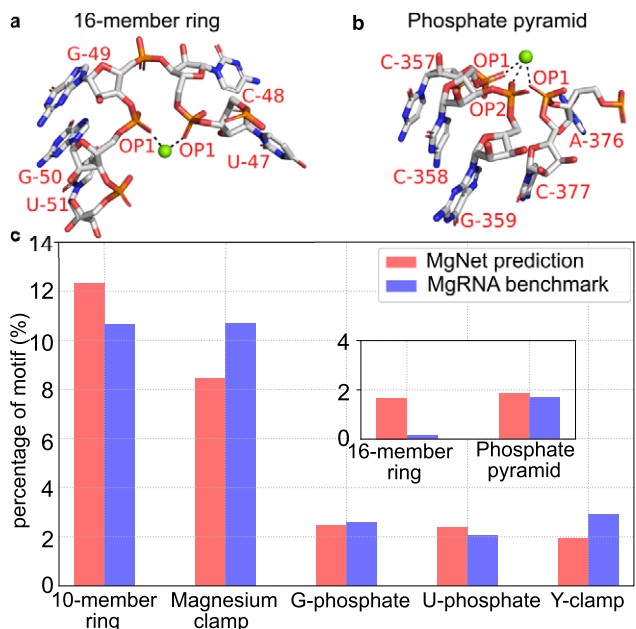

**Fig. 5.** Representative sites for newly discovered motifs and relative abundance of various motifs. (*a,b*) Representative sites are defined by PDB codes, chain id, and the predicted Mg²⁺ residue number as follows: (*a*) "16-member ring" (1QU2-T-9) and (*b*) "Phosphate pyramid" (4FAR-A- 30). Magnesium ions and inner-sphere interactions are shown in green spheres and black dashed lines, respectively. The coordinating RNA atoms and nearby nucleotides are labelled with red text. The "16-member ring" motif involves two inner-sphere coordinating oxygen atoms from two phosphate groups, respectively, separated by one residue (not consecutive phosphate groups). The two coordinating oxygen atoms, the RNA backbone atoms in between, and the Mg²⁺ form a ring with 16 atoms. The "Phosphate pyramid" motif contains either a "10-member ring" or a "16-member ring" with another inner-sphere ion coordinating the phosphate oxygen atoms, forming a triangular pyramid. (*c*) Relative abundance of the top-5 previously reported and newly discovered inner-sphere Mg²⁺ binding motifs in general set (red) and MgRNA benchmark set (Zheng *et al.*, 2015) (blue). The two newly discovered motifs are shown in the inset. The percentage of each motif is calculated by dividing the number of the sites belonging to the corresponding motif by the total number of sites with inner-sphere coordinating RNA atoms.

bar graph shows that the "Magnesium clamp" and the "10-member ring" motifs are the top-2 abundant motifs in both the general set and the MgRNA benchmark set, and the "G-phosphate", "U-phosphate", and "Y-clamp" motifs occur at similar levels of abundance. The newly discovered motifs are shown in the inset of the figure. The similar abundance of the "Phosphate pyramid" motif for both the general set and the MgRNA benchmark set indicates that this new motif is already in the MgRNA benchmark set and was probably overlooked in the previous study (Zheng *et al.*, 2015). Interestingly, the abundance of the "16-member ring" motif in MgRNA benchmark set is significantly lower than that in the general set. By investigating the sites that are identified as a "16-member ring" motif in the general set, we find that 65% of the sites belong to structures not included in the MgRNA benchmark set. We have also examined the corresponding experimental structures for the 21 and 20 predicted Mg²⁺ sites in the "Phosphate pyramid" and the "16-member ring" motifs, respectively, and found that the MgNet predictions are consistent with experimental results. Specifically, 17 and 13 predicted sites of the "Phosphate pyramid" and the "16-member ring" motifs have the corresponding experimentally observed ion binding sites, which constitute around 80.95 and 65.00% of the total predicted sites, respectively. The remaining predicted sites are either those without corresponding experimental ions or with ions other than Mg²⁺. The possible reason for these sites with missing experimental counterparts could be the quality of the dataset (i.e. ions that could exist in the structures but be overlooked by experiments). For this reason, although these motifs are discovered by our machine-learning model, further computational and experimental studies would be desirable to validate these newly identified motifs in RNA-Mg²⁺ interactions.

## Discussion

MgNet is a machine-learning method that uses a deep learning graphical convolutional neural network to predict Mg²⁺ binding sites for a given RNA structure. Currently, the model is trained to

predict $Mg^{2+}$ binding sites. With the increasing number of known RNA structures, we can realistically expect that the accuracy of MgNet predictions will continuously improve. Furthermore, with the increasing availability of nucleic acid structures with different types of bound ions, we can expect the extension of the applicability of the method for other metal ions and other nucleic acids (DNAs).

Comparisons with other existing approaches such as MetalionRNA (Philips *et al.,* 2011), MD simulations (Fischer *et al.,* 2018), and Brownian dynamics simulations (Hermann and Westhof, 1998) indicate that MgNet can lead to notable improvements in the prediction accuracy for $Mg^{2+}$ binding sites. Furthermore, saliency map analysis identifies and visualizes the RNA atoms that are most critical for $Mg^{2+}$ binding, and the information can facilitate our understanding of metal ion-RNA interactions. In contrast to physics-based models, which are usually excessively demanding in computational and human resources, with 3D RNA structures as the input and the predicted metal ion binding sites as the output, MgNet here can be conveniently implemented as a computationally efficient module that can be readily integrated into any automated processes.

**Open Peer Review.** To view the open peer review materials for this article, please visit http://doi.org/10.1017/qrd.2022.17.

**Acknowledgements.** We thank Prof. Jianlin Cheng for helpful discussions and Dr. Travis Hurst for the critical reading of the manuscript.

**Supplementary materials.** To view supplementary material for this article, please visit http://doi.org/10.1017/qrd.2022.17.

**Data availability statement.** The data supporting the findings of this study are available in the manuscript or in the supplementary materials.

**Code availability statement.** The source code can be downloaded from: https://github.com/Vfold-RNA/MgNet. The associated documentation is also available on the GitHub page.

**Author contributions.** Y.Z. and S-J.C. conceived the project. S-J.C. supervised the project. Y.Z. performed the data analysis, machine learning, and interpreted the data. Y.Z. and S-J.C. wrote the manuscript. All authors have read and approved the manuscript.

**Financial support.** This work was supported by the National Institutes of Health under Grant R35-GM134919 to S-J. C.

**Conflicts of interest.** The authors declare no conflicts of interest.

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
