## [Reviewer Report]

*Comments to Author*: Correct modelling of RNA structure is imperative for the studies of RNA-based medicines, RNA interactions with proteins, etc. In the manuscript the authors try to improve the modelling of the integral yet elusive component of RNA structure such as the binding of Mg2+ ions using a deep learning approach.

I find the manuscript interesting, highly relevant to the today’s challenges in RNA modelling, and overall well-written.

My major comment is the lack of a user-friendly tutorial/documentation supporting the GitHub code for MgNet, which limits the usefulness of the approach and is a pity! This tutorial/documentation should either be provided on the GitHub page or/and presented as a supplementary file in the paper.

Minor comments:

I find this sentence somewhat incomprehensible:

Page 1 lines 63-66. An flexible RNA can lead to an ensemble of low-energy conformations, and Mg2+ binding preferences may be different in different conformations, and may induce the conformational change of the target RNA (Bergonzo and Cheatham, [4]; Bergonzo et al., 2015, [5]).

You start by saying that experimental studies of Mg-RNA interactions are difficult and then provide a conclusion derived by three computational studies. Please paraphrase.

Page 5, in the beginning of Methods section, the authors say “we remove redundant structures of the same RNA”. Please clarify what do you mean by that. Do you remove RNA structures with same sequences, same set of 3D elements (e.g. hairpin, bulge, etc.)? Also please clarify what do you mean by “similar Mg2+ binding sites”, maybe you should use some measure of RMSD of Mg2+ ions? Also, if you have identified same RNA structures with sufficiently different (let’s say RMSD > 2-3Å) binding sites for the Mg2+ ions, it should be commented on.

Also, did you use any criteria or some randomising procedure when dividing the 177 RNA structures into the 5 sets or contrary collected RNA structures with similar sequences/3D motifs/MG2+ binding sites in one group?

Fig.1 Panel b left hand-side image, by looking at a “hive” of the hot spots for Mg2+ binding predicted by MgNet which surrounds an RNA molecule, it seems to me that Mg2+ can bind practically anywhere. According to the provided description of the method, MgNet output the probabilities of the Mg2+ binding. I presume that the hot spots density should be higher in the regions where the binding of an ion should be most probable. Can this be integrated into an image, through some sort of shading? Also, if you just go from the 3D binding probability densities, why do you need clustering? Isn’t it redundant? Please comment on that.

Page 7. Typo, line 197 “experimental RNA structure, where many experimentally sites not included in the set could be…”

I believe the authors meant to write "experimentally derived" or something similar?

Fig. 2 panel A. Some variation of the success rate is seen depending on the tested set (one out of five). Can you comment on that? See my question above about the division into 5 sets. According to your data, is there some RNA structural motif that appears to be more difficult to provide a prediction for? It could be interesting to discuss these aspects.

---

## [Reviewer Report]

*Comments to Author*: In this article, the authors present a machine-learning (ML) approach, called MgNet, to predict Mg2+ binding sites in RNA molecules. This is an important topic, since it is often difficult to observe Mg2+ ions though cryo-EM techniques. The paper is very well presented and the ML approach is validated over a large number of Mg2+-containing structures. MgNet is based on network theory and is an interesting innovation with respect to knowledge-based methods (e.g. Metalion) and molecular dynamics approaches. In my view, the paper will be of broad interest for the RNA community, and useful for structural biologists using cryo-EM.

The paper requires a couple of minor revisions. The prediction of Mg2+ ions has been has made extensive use of quantum mechanical methods, an aspect that should be discussed and is missing in the current version of the paper. The authors claim that this approach can be used by structural biologists to predict metal binding sites. A. github link to the code is provided, but its documentation appears of difficult understanding for an audience that goes beyond computational scientists. This is an issue that should be addressed, with clear guidance in the paper.

---

## [Reviewer Report]

*Comments to Author*: Reviewer #1: In this article, the authors present a machine-learning (ML) approach, called MgNet, to predict Mg2+ binding sites in RNA molecules. This is an important topic, since it is often difficult to observe Mg2+ ions though cryo-EM techniques. The paper is very well presented and the ML approach is validated over a large number of Mg2+-containing structures. MgNet is based on network theory and is an interesting innovation with respect to knowledge-based methods (e.g. Metalion) and molecular dynamics approaches. In my view, the paper will be of broad interest for the RNA community, and useful for structural biologists using cryo-EM.

The paper requires a couple of minor revisions. The prediction of Mg2+ ions has been has made extensive use of quantum mechanical methods, an aspect that should be discussed and is missing in the current version of the paper. The authors claim that this approach can be used by structural biologists to predict metal binding sites. A. github link to the code is provided, but its documentation appears of difficult understanding for an audience that goes beyond computational scientists. This is an issue that should be addressed, with clear guidance in the paper.

Reviewer #2: Correct modelling of RNA structure is imperative for the studies of RNA-based medicines, RNA interactions with proteins, etc. In the manuscript the authors try to improve the modelling of the integral yet elusive component of RNA structure such as the binding of Mg2+ ions using a deep learning approach.

I find the manuscript interesting, highly relevant to the today’s challenges in RNA modelling, and overall well-written.

My major comment is the lack of a user-friendly tutorial/documentation supporting the GitHub code for MgNet, which limits the usefulness of the approach and is a pity! This tutorial/documentation should either be provided on the GitHub page or/and presented as a supplementary file in the paper.

Minor comments:

I find this sentence somewhat incomprehensible:

Page 1 lines 63-66. An flexible RNA can lead to an ensemble of low-energy conformations, and Mg2+ binding preferences may be different in different conformations, and may induce the conformational change of the target RNA (Bergonzo and Cheatham, [4]; Bergonzo et al., 2015, [5]).

You start by saying that experimental studies of Mg-RNA interactions are difficult and then provide a conclusion derived by three computational studies. Please paraphrase.

Page 5, in the beginning of Methods section, the authors say “we remove redundant structures of the same RNA”. Please clarify what do you mean by that. Do you remove RNA structures with same sequences, same set of 3D elements (e.g. hairpin, bulge, etc.)? Also please clarify what do you mean by “similar Mg2+ binding sites”, maybe you should use some measure of RMSD of Mg2+ ions? Also, if you have identified same RNA structures with sufficiently different (let’s say RMSD > 2-3Å) binding sites for the Mg2+ ions, it should be commented on.

Also, did you use any criteria or some randomising procedure when dividing the 177 RNA structures into the 5 sets or contrary collected RNA structures with similar sequences/3D motifs/MG2+ binding sites in one group?

Fig.1 Panel b left hand-side image, by looking at a “hive” of the hot spots for Mg2+ binding predicted by MgNet which surrounds an RNA molecule, it seems to me that Mg2+ can bind practically anywhere. According to the provided description of the method, MgNet output the probabilities of the Mg2+ binding. I presume that the hot spots density should be higher in the regions where the binding of an ion should be most probable. Can this be integrated into an image, through some sort of shading? Also, if you just go from the 3D binding probability densities, why do you need clustering? Isn’t it redundant? Please comment on that.

Page 7. Typo, line 197 “experimental RNA structure, where many experimentally sites not included in the set could be…“

I believe the authors meant to write “experimentally derived” or something similar?

Fig. 2 panel A. Some variation of the success rate is seen depending on the tested set (one out of five). Can you comment on that? See my question above about the division into 5 sets. According to your data, is there some RNA structural motif that appears to be more difficult to provide a prediction for? It could be interesting to discuss these aspects.